METHODS

# Bridging cancer cell-intrinsic driver genes and -extrinsic cell-cell communication with Driver2Comm

**Runzhi Xie, Junping Li, Yuxuan Hu** ⓘ *, **Lin Gao** ⓘ *

School of Computer Science and Technology, Xidian University, Xi'an, Shaanxi, China

* lgao@mail.xidian.edu.cn (LG); huyuxuan@xidian.edu.cn (YH)

## Abstract

Tumor development and progression are affected not only by cancer cell-intrinsic factors comprising complex genetic variations, but also by -extrinsic factors such as cell-cell communication (CCC)-mediated immunosuppression. However, whether and how these two types of factors influence each other remains an open question. We present Driver2Comm, a general computational framework designed to systematically identify intrinsic-extrinsic (IE) pathways that functionally connect cancer cell driver genes with their associated CCC signatures in the tumor microenvironment (TME). By applying Driver2Comm to single-cell and spatial transcriptomic datasets of multiple cancer types, we find that driver gene-associated CCC signatures play critical roles in immune regulation, metastasis, and therapy response. These signatures not only illuminate mechanisms of TME remodeling but also demonstrate clinical value in predicting patient survival and response to immune checkpoint blockade. Furthermore, Driver2Comm captures higher-order, cell-type-pair-specific CCC functional modules and spatially coherent CCC patterns in tissue contexts. As a generalizable tool, Driver2Comm bridges cancer genomics and cellular ecosystems, offering insights into biomarker discovery and combination therapy strategies.

## Author summary

In this study, we introduce Driver2Comm, a computational tool designed to bridge two critical aspects of cancer biology: intrinsic genetic drivers and extrinsic cellular communication. It is well known that cancer progression is influenced not only by mutations within cancer cells but also by interactions between different cell types in the tumor microenvironment. However, how these two layers connect has remained largely unclear. Using single-cell and spatial transcriptomic data, Driver2Comm enables the identification of potential signaling pathways linking cancer driver genes to their associated cell-cell communication events. We applied Driver2Comm across multiple cancer types, including pancreatic cancer, breast cancer, neuroblastoma, and oral squamous cell carcinoma, and

**Data availability statement:** All data set analyzed in this article are publicly available. Detail information has been shown in S1 - S5 Tables. Specifically, snRNA-seq data of PDAC can be downloaded from Gene Expression Omnibus (GEO) with accession number of GSE202051. ScRNA-seq data of breast cancer from Wu et al. can be download from GEO with accession number of GSE176078. ScRNA-seq data of breast cancer from Bassez et al. can be download from https://lambrechtslab.sites.vib.be/en/single-cell with data accession no. EGAD00001006608. Spatial transcriptomics data of OSCC can be download from GEO with accession number of GSE208253. ScRNA-seq data of neuroblastoma can be download from https://cellxgene.cziscience.com/collections/cee845e3-ec04-4781-9e2a-28734bb4f7ba. Bulk expression matrices and clinical information of the METABRIC cohort45 used in this study are available at https://www.cbioportal.org/study/summary?id=brca_metabric. Bulk RNA-seq data, copy number alterations and clinical information of PDAC from TCGA cohort [108] can be download from https://www.cbioportal.org/study/summary?id=paad_tcga_gdc A Python implementation for Driver2Comm and example scripts for associating driver genes of cancer cells with CCC in the TME using single-cell transcriptomics data are available at https://github.com/huBioinfo/Driver2Comm.

**Funding:** This work was supported by a National Natural Science Foundation of China (NSFC) grant No. 62422211, a Scientific Research Innovation Capability Support Project for Young Faculty No. SRICSPYF-ZY2025003 to YH, and NSFC grants No. 62550005, No. 62132015, and No. U22A2037 to LG. The funders had no role in study design, data collection and analysis, decision to publish, or preparation of the manuscript.

**Competing interests:** The authors have declared that no competing interests exist.

discovered genetic driver-associated cellular communication signatures tied to key processes such as immune evasion, metastasis, and treatment response. These signatures also showed clinical value in predicting patient survival and immunotherapy outcomes. By making these connections visible and interpretable, our work provides a new way to understand how tumors function as complex ecosystems, offering potential clues for future combination therapies.

## Introduction

Tumor is a complex cellular ecosystem and its initiation and progression can be attributed to cancer cell-intrinsic and -extrinsic factors [1,2]. The intrinsic factors are mainly composed of genetic and epigenetic events occurring in cancer cells, such as driver gene mutations, structure variations and epigenetic modifications that result in the silencing of tumor suppressor genes or activation of oncogenes [3]. A key extrinsic factor comes from cell-cell communication (CCC) in the tumor microenvironment (TME), which has been demonstrated closely associated with tumorigenesis, tumor immune evasion, tumor metastasis, and therapy response [4–6]. The most famous example is the tumor immune evasion caused by cancer cell-T cell communication via $CD274 - PDCD1$ (also known as PD-L1 – PD1) interaction [7]. Interactions between these two types of factors have been experimentally investigated in previous studies [1,8,9]. Accumulating evidence indicates that oncogenic driver genes can remodel CCC in the TME to facilitate immune evasion. For example, $EGFR$-mutant non-small-cell lung cancer accomplishes immune evasion through inducing activation of the PD-L1 – PD1 pathway [10,11]. Similarly, oncogenic $KRAS$ mutation promotes an immune-suppressive TME in colorectal cancer and leads to resistance to anti-PD1 therapy by activating the $CXCL3 - CXCR2$ signaling pathway between cancer cells and myeloid-derived suppressor cells [12]. Conversely, extrinsic CCC signals can also shape the selective landscape of cancer evolution, influencing which genetic alterations are retained and clonally expanded. Specifically, immune cell-mediated interactions, such as IFN-γ signaling, PD-L1 engagement, or antigen presentation, can eliminate immunogenic subclones while favoring those harboring mutations that enable immune escape [13–15]. These observations imply that extrinsic communication signals not only respond to, but also shape, the selection of driver gene alterations. However, the mechanisms of how the cancer cell-intrinsic driver genes interact with -extrinsic CCC events in the TME are yet to be fully elucidated in most cancer types.

There are several well-known experimental models for investigating the influence of cancer cell-intrinsic driver genes on -extrinsic CCC in the TME, such as the genetically engineered mouse model (GEMM), the patient-derived xenograft (PDX) model and the organotypic tumor spheroid. GEMM can mimic the TME under controlled genetic alteration conditions, providing a tractable and accurate test of validation of hypothesis related to tumor in vivo [16]. PDX models closely recapitulate the heterogeneity and complexity of human tumors [17–19]. Organotypic tumor spheroids allow

PLOS Computational Biology

for the study of cell-cell interactions and drug responses in a three-dimensional context that mimics the architecture and microenvironment of tumors [20–22]. While these experimental models have achieved significant success, their use could be time-consuming, labor-intensive and costly. With the emergence of single-cell transcriptomics technologies, computational methods have been developed for inferring CCC in the tissue microenvironment [23–34] and some of them can be used to detect condition-specific CCC. Kumar et al. identified ligand-receptor pairs potentially associated with different patient outcomes using regression models [23]. MultiNicheNet [28] and LIANA+ [32] inferred differential expressed ligand-receptor pairs between conditions. However, none of these computational methods explicitly treated the driver genes of cancer cells as conditions to link them with CCC pathways in the TME.

For systematic investigation of the interplay between cancer cell-intrinsic and -extrinsic factors, we develop a general computational framework, Driver2Comm, for identifying intrinsic-extrinsic (IE) pathways that functionally connect driver genes in cancer cells with their associated CCC signatures in the TME using single-cell transcriptomics data. Inspired by Genome-Wide Association Studies (GWAS) [35], Driver2Comm uses hypothesis testing to identify cancer driver gene-associated CCC signatures, which are further evaluated through four strategies including visualization and functional analysis of IE pathways, independent tumor cohort-based survival analysis, and immune-related score-based benchmarking. We validate the utility of Driver2Comm across multiple cancer types, showing its ability to reveal IE pathways involved in immune suppression, metastasis, and therapy resistance, and further demonstrate its extension to spatial transcriptomics and multi-cell-type ecosystem analysis. By bridging cancer genomics and cellular communications, Driver2Comm represents a promising tool for developing novel combination therapies of targeted therapy and immunotherapy. A software package implementing the Driver2Comm has been deposited at https://github.com/huBioinfo/Driver2Comm.

## Results

### Overview of Driver2Comm

Driver2Comm is designed for associating cancer cell-intrinsic driver genes with -extrinsic CCC in the TME (Fig 1a). It requires single-cell transcriptomics data of tumor tissues with both cell type and cancer cell driver gene annotations as inputs. First, Driver2Comm uses our previously developed CytoTalk [36], a CCC analysis tool, to construct a multi-cell-type communication (MCTC) network *de novo* for each tumor sample, which includes both intracellular and intercellular gene interactions (Methods). In this study, we aimed to link driver genes of cancer cells (TC) to CCC-regulated immune responses in the TME and thus chose macrophages (Mφ) and CD8+T cell (CTL) types to construct the MCTC networks because of their important roles in innate and adaptive immunity, respectively. It is worth noting that more immune and stromal cell types of interest can be incorporated in the MCTC network construction.

Next, Driver2Comm identifies frequent subnetworks within the MCTC networks derived from all tumor samples as robust cancer cell-extrinsic CCC signatures, which are used together with cancer cell-intrinsic driver genes as two factors for following association testing. Inspired by the GWAS, Driver2Comm uses the Fisher's exact test to determine the significance of the association between the two factors and identify cancer driver gene-associated CCC signatures. To explicitly build a bridge between cancer cell-intrinsic and -extrinsic factors, we define an IE pathway as the shortest path between a driver gene and its associated CCC signature in the MCTC network (Methods). We further propose a downstream evaluation framework to provide biological insights into the relationship between cancer cell-intrinsic and -extrinsic factors, including functional enrichment analysis of IE pathways, and CCC signature-based survival analysis and immune-related scoring (Fig 1b; Methods).

### Driver gene-associated CCC signatures in pancreatic ductal adenocarcinoma

To demonstrate the utility of Driver2Comm, we first applied it to a single-cell transcriptomics dataset of pancreatic ductal adenocarcinoma (PDAC) [37]. Since driver gene annotations of each patient were not provided in this dataset, we first

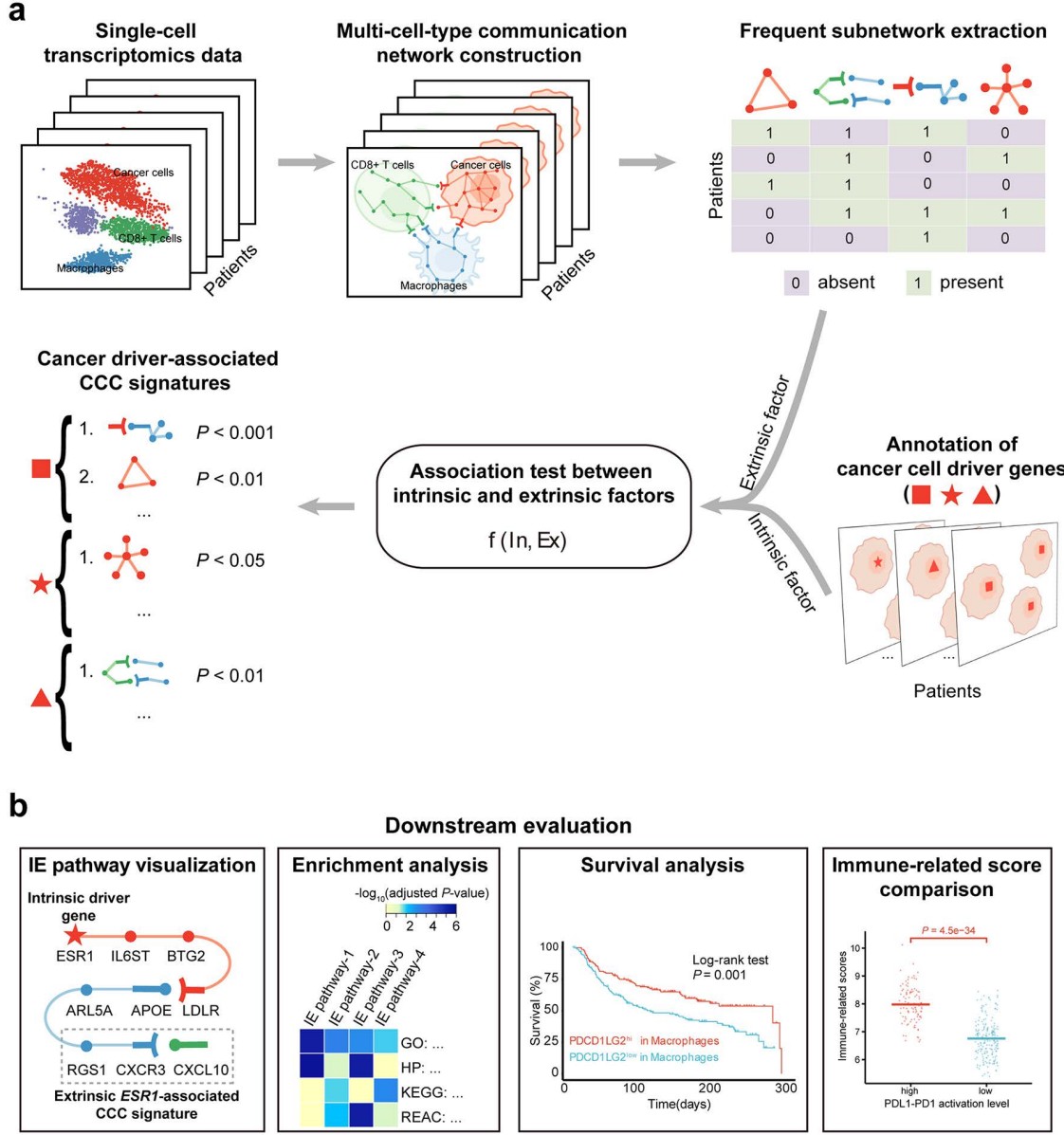

**Fig 1. Schematic overview of the Driver2Comm framework. a Driver2Comm requires single-cell transcriptomics data of tumor tissues with cell types and cancer cell driver gene annotations as inputs.** A multi-cell-type communication (MCTC) network among cancer cells, CD8+ T cells, and macrophages is first constructed based on the CytoTalk algorithm (step 1). Frequent subnetworks are extracted as robust signatures of MCTC network (step 2). Both frequent subnetworks and cancer driver gene of each patient formed as a binary matrix using one-hot encoding. Finally, frequent subnetwork encoding vectors representing extrinsic factors of cancer cells and cancer driver genes encoding vectors representing intrinsic factors are used as input to association test function $f(In, Ex)$, where $In$ and $Ex$ are one-hot encoding vectors of the cancer driver gene and frequent subnetworks, respectively. Fisher's exact test is used as the association test function in this study. Frequent subnetworks which are significantly associated with cancer driver gene (adjust P value <0.05) called cancer driver gene-associated CCC signatures are ranked according to the adjust P value (step 3). **b** Downstream evaluation of Driver2Comm results. Driver2Comm suggests four strategies to evaluate the biological functions of these CCC signatures, including IE pathways visualization, functional enrichment analysis, survival analysis, and immune-related score comparison.

used inferCNV [38] to identify variated driver genes of each patient (Methods; Fig A in S1 Text). Next, we applied Driver-2Comm to establish the association between cancer cell-intrinsic driver genes and -extrinsic CCC among cancer cells, macrophages and CD8+ T cells in the TME. Driver2Comm identified CCC signatures significantly associated with *BRCA1*, *AKT2*, *CDKN2A*, *GATA6*, *PTEN*, and *MET* (Fig B in S1 Text). We found that the number of CCC signatures associated with different driver genes varies, reflecting the distinct degree of influence of drivers on CCC (Fig 2a). We selected the top two groups of CCC signatures in amount, those associated with *BRCA1* and *AKT2*, as representatives for the subsequent analysis.

We first visualized the association of *BRCA1* and *AKT2* with their corresponding CCC signatures, as well as the top ten enriched biological functions of these signatures (Fig 2b). As shown, the key CCC signatures associated with *BRCA1* are *HGF* (Mφ) – *SDC1* (TC), *HGF* (Mφ) – *MET* (TC), *HGF* (Mφ) – *ST14* (TC), and *CCL5* (CTL) – *SDC1* (TC). These CCC signatures were enriched in functions such as chemotaxis, regulation of cell adhesion, inflammatory response, and immune system process, suggesting that *BRCA1* may regulate tumor invasion, immune response, and TME remodeling through these CCC pathways, thereby facilitating immune evasion. In contrast, *AKT2* is primarily associated with *THBS1* (Mφ) – *ITGA3* (TC), *THBS1* (Mφ) – *ITGA6* (TC), *THBS1* (Mφ) – *SDC1* (TC), and *SPP1* (Mφ) – *ITGA4* (CTL), with these signatures enriched in functions like cell migration, locomotion, and regulation of cell motility, suggesting that *AKT2* primarily drive tumor invasion and metastasis through *THBS1* and integrin family molecules. We also visualized the IE pathways of *BRCA1* and *AKT2* to their corresponding key CCC signatures (Figs 2c, 2d; Figs C and D in S1 Text). Interestingly, both IE pathways involve multiple associated pathways, suggesting that driver genes can regulate CCC through different routes. Overall, *BRCA1*-associated CCC signatures are more focused on immune and inflammatory regulation within the TME, which represent potential immunotherapy targets in PDAC. On the other hand, *AKT2*-associated CCC signatures are more involved in tumor invasion and metastasis, making them serve as promising candidate targets for inhibiting PDAC progression.

Furthermore, we investigated the prognostic effects of the CCC signatures associated with *BRCA1* and *AKT2*. To this end, we performed survival analysis using the bulk RNA-seq data from The Cancer Genome Atlas (TCGA). By deconvolution of the bulk transcriptomics data, we evaluated the prognostic values of *BRCA1* and *AKT2*-associated CCC signatures (Methods; Fig E in S1 Text). The results revealed that some CCC signatures associated with *BRCA1* and *AKT2* had prognostic value in patients with the corresponding driver. Specifically, six *BRCA1*-associated CCC signatures showed significant prognostic effects in the *BRCA1*-driven TGCA patients (Fig F in S1 Text). Notably, CCC signatures involving *HLA-A* exhibited significant prognostic effects in *BRCA1*-driven PDAC patients, with co-silencing of *HLA-A* (CTL) – *APLP2* (Mφ), *HLA-A* (CTL) – *ERBB2* (TC), and *HLA-A* (CTL) – *LILRB1* (Mφ) leading to the poorest patient survival (Fig 2e). Additionally, we found that eight *AKT2*-associated CCC signatures had significant prognostic values in the *AKT2*-driven PDAC patients (Fig G in S1 Text). Specifically, CCC signature related to genes in the integrin family showed significant prognostic effects in *AKT2*-driven PDAC, with high expression of genes in *SPP1* (Mφ) – *ITGA4* (CTL) associated with the most unfavorable clinical outcomes. In contrast, low expression of *ITGA3* in CCC signature 10 was linked to a better prognosis, while *ITGA3*low (TC) – *THBS1*hi (Mφ) – *SDC1*low (TC) correlated with the best prognosis (Fig 2f). Taken together, these results demonstrated that Driver2Comm can successfully identify distinct pathways reshaping the TME in PDAC and detect CCC signatures with driver-specific prognostic effects.

## Driver gene-associated CCC signatures in breast cancer

To further investigate the utility of Driver2Comm in different types of tumors, we next applied it to a single-cell transcriptomics dataset of human breast cancer [39]. The dataset contained three common subtypes of breast cancer, which are ER+, HER2+, and TNBC subtypes. Specifically, the driver gene for ER+ subtypes is *ESR1*, and for HER2+ subtype is *ERBB2*. We identified 15 and 3 *ESR1*- and *ERBB2*-associated CCC signatures, respectively (Fig H in S1 Text). Similar to PDAC, we observed a differing number of CCC signatures associated with driver genes (Fig 3a). We then selected the

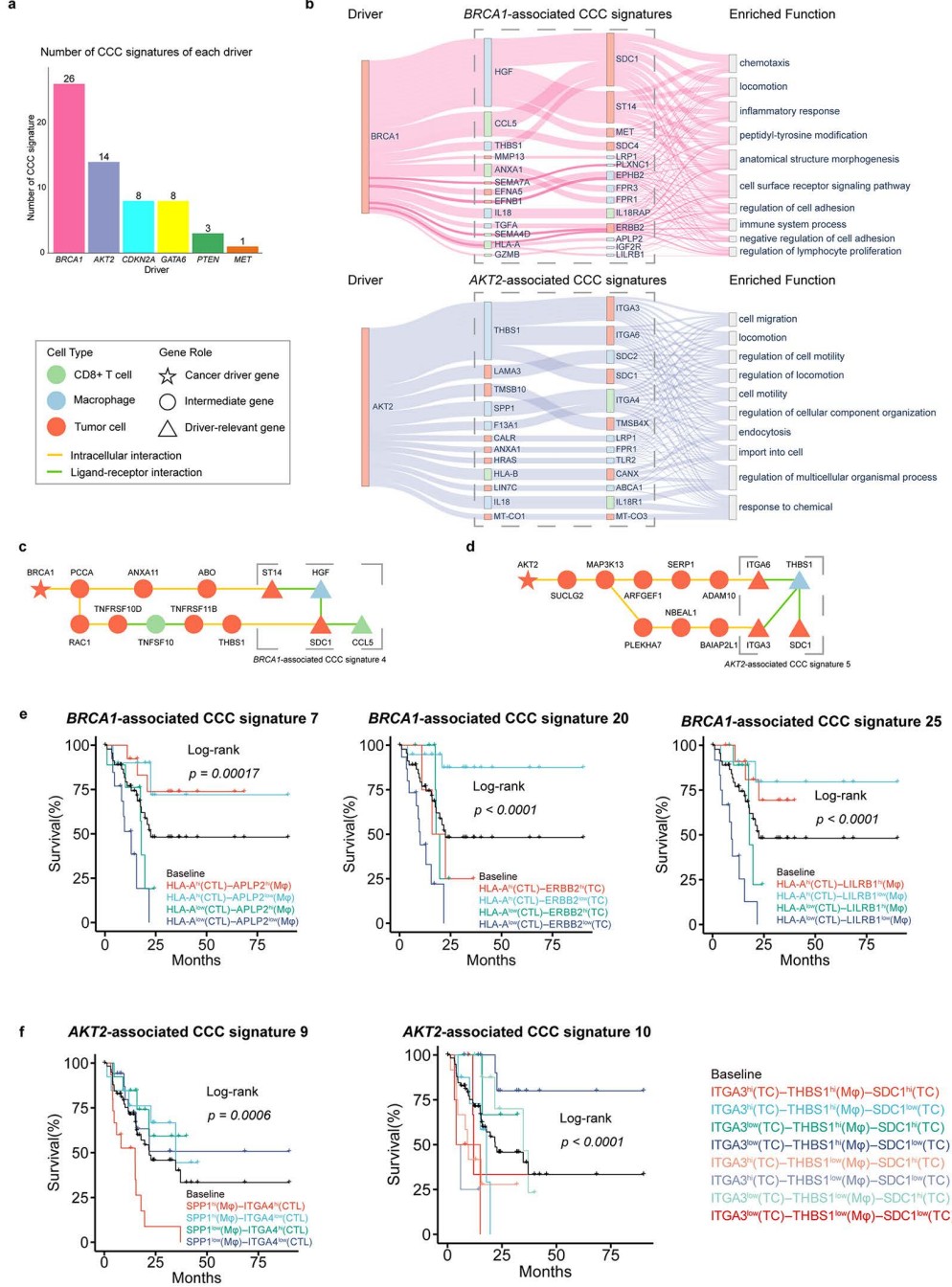

**Fig 2. Driver gene-associated CCC signatures in the PDAC dataset. a The number of CCC signatures associated with drivers in the PDAC dataset. b Sankey diagram describing the correlation among driver genes, driver gene-associated CCC signatures and the corresponding enriched functions.** For nodes in driver, and driver gene-associated CCC signatures, node labels are gene names and different colors are used to distinguish cell types with red for tumor cells, blue for macrophages, and green for CD8+T cells. For nodes in enriched function, node labels are top 10 enriched function of genes in CCC signatures. The line width reflected the statistical significance of CCC signature enriched in the given functional category, as determined by Gene Ontology (GO) enrichment analysis (adjusted *P* value calculated using the Benjamini-Hochberg procedure). It is inversely proportional to the adjusted *P* value. **c** IE pathway between *BRCA1* and *BRCA1*-associated CCC signature 4. **d** IE pathway between *AKT2* and *AKT2*-associated CCC signature 5. **e** Kaplan-Meier analysis of survival outcomes in *BRCA1* driven TCGA cohort with different expression level of the *BRCA1*-associated CCC signatures. **f** Kaplan-Meier analysis of survival outcomes in *AKT2* driven TCGA cohort with different expression level of the *AKT2*-associated CCC signatures. Patients were stratified according to the expression of genes involved in driver gene-associated CCC signatures.

For each gene in a given CCC signature, patients were categorized into highly-expressed and lowly-expressed groups based on the median expression value. All possible combinations of high/low states across the involved genes in the CCC signatures were considered, and each colored curve represents one such combinatorial expression pattern, as indicated in the legend. Survival differences among all stratified groups were evaluated using a multi-group log-rank test, and the reported *P* values correspond to the global comparison across all groups within each panel. TC, Tumor Cell; CTL, CD8+T cell; Mφ, Macrophage.

more abundant *ESR1*-associated CCC signatures for subsequent analysis. We first visualized the association of *ESR1* with *ESR1*-associated CCC signatures, as well as the top ten enriched biological functions of these signatures (Fig 3b). The results showed that the *ESR1*-associated CCC signatures were enriched with functions related to tumor invasion and metastasis, including locomotion, cell migration, and cell motility. We next focused on *ESR1*-associated IE pathways. Notably, we found that *IL6ST* was identified in all predicted IE pathways, suggesting its strong relationship with both *ESR1* and tumor immunity (Fig 3c and Fig I in S1 Text). Previous studies have shown that *IL6ST* is linked to proliferation in ER+ breast cancer [40,41]. Besides, *IL6ST* also drives multiple cancer cell-extrinsic pro-tumor activities in the TME [42,43], such as inducing the expression of *VEGFA*, which promotes angiogenesis and metastasis of breast cancer (Fig 3d) [44]. To further investigate the functions of *ESR1*-involved IE pathways, we conducted enrichment analysis of these pathways. We found that they were enriched for tumor immunity-related functions such as apoptotic, immune system process and inflammatory response. In addition, several *ESR1*-associated IE pathways were enriched for genes in the pathways in cancer curated in the KEGG database (Fig 3e).

To further evaluate the prognostic effect of *ESR1*-associated CCC signatures identified by Driver2Comm, we performed survival analysis using an independent breast cancer dataset, which includes bulk transcriptomics data profiled from the METABRIC cohort [45] (Methods).. We found that 10 *ESR1*-associated CCC signatures had significant prognostic effects in the METABRIC ER+ patients (Fig J in S1 Text) and five of them were specifically associated with clinical outcomes of ER+ patients, including *ADCY1* (TC) – *GNAI2* (Mφ), *SDC4* (TC) – *CXCL10* (Mφ) – *CXCR3* (CTL), *NRP2* (Mφ) – *SEMA3B* (TC), *APOE* (Mφ) – *LRP2* (TC) and *CXCL10* (Mφ) – *CXCR3* (CTL) interactions. It is worth noting that four of them were ranked at the top based on association tests (Fig H in S1 Text), highlighting the ability of Driver2Comm to prioritize *ESR1*-associated CCC signatures with ER+ specific prognostic values.

We further found two distinct prognostic patterns with respect to gene expression levels of the *ESR1*-associated CCC signatures. One pattern was characterized by all genes in the signature with similar expression levels leading to the poorest patient survival. For example, high expression of *SDC4* (TC), *CXCL10* (Mφ), and *CXCR3* (CTL) in Signature 3 (Fig 3f) and high expression of *TLR5* (Mφ) and *ZG16B* (TC) in Signature 8 (Fig 3g) were associated with the most unfavorable clinical outcomes of ER+ patients. Previous studies also demonstrated the pro-tumor role of the signaling axis *CXCL10* – *CXCR3* in breast cancer [46–50]. Moreover, Hilborn et al. showed that the expression of *CXCL10* and *CXCR3* could be a prognostic predictor of ER+ breast cancer patients treated with tamoxifen [51]. In contrast, another pattern described that genes with opposite expression levels in the signature were associated with the poorest patient survival. For instance, high expression of *NRP2* (Mφ) and low expression of *SEMA3B* (TC) in Signature 4 (Fig 3h) and high expression of *IL1B* (Mφ) and low expression of *ADRB2* (CTL) in Signature 10 (Fig 3i) were associated with the poorest outcomes in the patients with ER+ breast cancer. Consistent with our observations, it was reported that expression of *NRP2* in tumor-associated macrophages was associated with immunosuppressive and tumor-promoting microenvironment [52] and that *SEMA3B* functioned as a tumor suppressor in breast cancer [53].

Another interesting example is the expression of *VEGFA* (Mφ) and *GPC1* (TC) in Signature 9. We found that low expression of *VEGFA* (Mφ) and high expression of *GPC1* (TC) were associated with the most favorable outcomes, whereas low expression of *VEGFA* (Mφ) and low expression of *GPC1* (TC) in Signature 9 were linked to the poorest outcomes in the patients with ER+ breast cancer (Fig J in S1 Text). Notably, these two genes did not exhibit a significant prognostic effect individually (Fig K in S1 Text), emphasizing the prognostic capacity of driver gene-associated CCC signatures

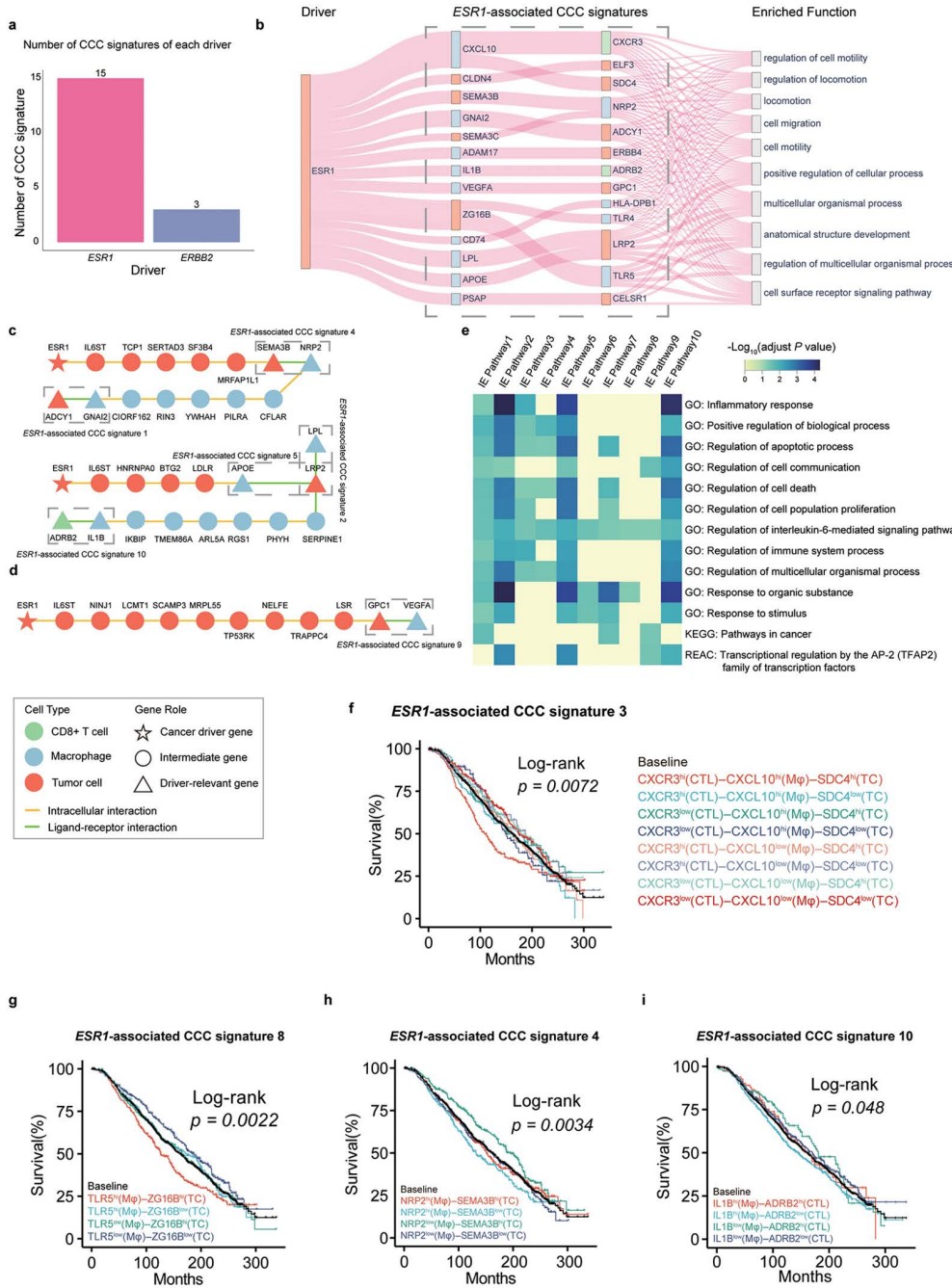

**Fig 3. Driver gene-associated CCC signatures in the breast cancer dataset from Wu et al. a** The number of CCC signatures associated with drivers in breast cancer dataset. **b** Sankey diagram describing the correlation among *ESR1*, *ESR1*-associated CCC signatures and the corresponding enriched functions. For nodes in driver, and driver gene-associated CCC signatures, node labels are gene names and different colors are used to distinguish cell types with red for tumor cells, blue for macrophages, and green for CD8+T cells. For nodes in enriched function, node labels are top 10 enriched function of genes in CCC signatures. The line width reflected the statistical significance of CCC signature enriched in the given functional category, as determined by Gene Ontology (GO) enrichment analysis (adjusted *P* value calculated using the Benjamini-Hochberg procedure). It is inversely proportional to the adjusted *P* value. **c** The IE pathways between *ESR1* and the top 2 *ESR1*-associated CCC signatures in the MCTC network. **d** IE pathway between *ESR1* and *ESR1*-associated CCC signature 9. **e** Genes on the top 10 IE pathways of *ESR1* are enriched for Gene Ontology (GO) biological processes, KEGG, and Reactome (REAC) database. Shades of blue in the heatmap are inversely proportional to the enrichment P values that were adjusted for multiple testing using the Benjamini-Hochberg method. Nonsignificant *P* values (> 0.05) are indicated in yellow. **f-i** Kaplan-Meier analysis of survival outcomes in ER⁺ cohort in METABRIC (n = 1196 patients in total) with different expression level of the *ESR1*-associated CCC

signature. Patients were stratified according to the expression of genes involved in *ESR1*-associated CCC signatures. For each gene in a given CCC signature, patients were categorized into highly-expressed and lowly-expressed groups based on the median expression value. All possible combinations of high/low states across the involved genes in the CCC signatures were considered, and each colored curve represents one such combinatorial expression pattern, as indicated in the legend. Survival differences among all stratified groups were evaluated using a multi-group log-rank test, and the reported *P* values correspond to the global comparison across all groups within each panel. TC, Tumor Cell; CTL, CD8+ T cell; Mφ, Macrophage.

as a whole. Besides, genes in the remaining *ESR1*-associated CCC signatures were also known to play important roles in tumor progression, metastasis, and patient survival prediction (Note A in S1 Text).

Taken together, these results demonstrated the important role of IE pathways in providing biological insights into the interaction mechanism between the intrinsic and extrinsic factors of cancer cells and the effectiveness of Driver2Comm in identifying novel CCC signatures with prognostic value in ER+ patients.

### *ESR1*-associated CCC signatures are linked to anti-PD1 therapy responses

To further evaluate the reliability and reproducibility of the CCC signatures identified in Wu et al., we applied Driver2Comm on another independent breast cancer dataset from Bassez et al. [54], which includes paired primary tumor biopsy samples from patients before and during anti-PD1 therapy. We focused on data from primary untreated biopsies and identified 22 *ESR1*-associated CCC signatures (Fig L in S1 Text). As expected, there were reasonable overlaps in the genes of the CCC signatures in both datasets (Fig L in S1 Text). Furthermore, we found that genes in the CCC signatures associated with the same intrinsic factors in both datasets were enriched for similar functions (Fig L in S1 Text). Specifically, among the top 10 overlapping enriched functions of *ESR1*-associated CCC signatures, we observed functions related to tumor metastasis, such as cell migration and regulation of cell motility, which suggested potential mechanisms underlying *ESR1*-driven breast tumor metastasis (Fig L in S1 Text).

Moreover, Bassez et al. simultaneously performed single-cell T cell receptor sequencing (scTCR-seq) on the treated tumor biopsy samples to assess whether patients responded to anti-PD1 therapy. Patients with clonotype expansion were categorized as response group ("E"), while those with limited or no clonotype expansion were categorized as non-response group ("NE"). Since the absence of HER2 + patients in the NE group, we focused on ER+ patients for this case study. We applied Drive2Comm on both groups of ER+ patients to determine whether it could help identify *ESR1*-associated CCC signatures or genes linked to anti-PD1 therapy response (Fig 4a).

The identified *ESR1*-associated CCC signatures included known immune checkpoint interaction: *NECTIN2* (TC) –*TIGIT* (CTL), *HLA-DQA2* (Mφ) – *LAG3* (CTL) (Fig 4b) [54,55]. We next investigated the biological functions of these CCC signatures. The enrichment analysis revealed that *ESR1*-associated CCC signatures reflected the distinct functional states of the TME in the two patient groups. Functions enriched in the NE group were more related to immune escape such as cell migration and cell-cell adhesion, while those in the E group were more related to enhancing the recognition and killing ability of immune cells, including cell killing, and cell recognition.

We further explored the NE-specific CCC signatures and found the most prominent interactions involving *ZG16B*(TC), which connected with *CXCR4* (Mφ), *TLR2* (Mφ), *TLR4* (Mφ), *TLR5* (Mφ), and *TLR6* (Mφ). Previous studies have demonstrated the pro-tumor effect of these interactions in the TME [56–58]. Furthermore, *ZG16B* has recently been identified as a druggable pro-tumorigenic target in breast cell transformation [59]. However, the correlation of *ZG16B* to immunotherapy and its role in *ESR1*-driven breast cancer has not been explored. We further investigated the effect of *ZG16B* in *ESR1*-driven breast cancer. UMAP visualization of *ESR1*-driven breast tumor cells revealed significantly higher *ZG16B* expression in the NE group compared to the E group, indicating that *ZG16B* overexpression is associated with anti-PD1 therapy response (Fig 4c and 4d). We then evaluated the prognostic effect of *ZG16B*(TC) in *ESR1*-driven breast cancer using METABRIC data. Survival analysis showed that high expression of *ZG16B*(TC)

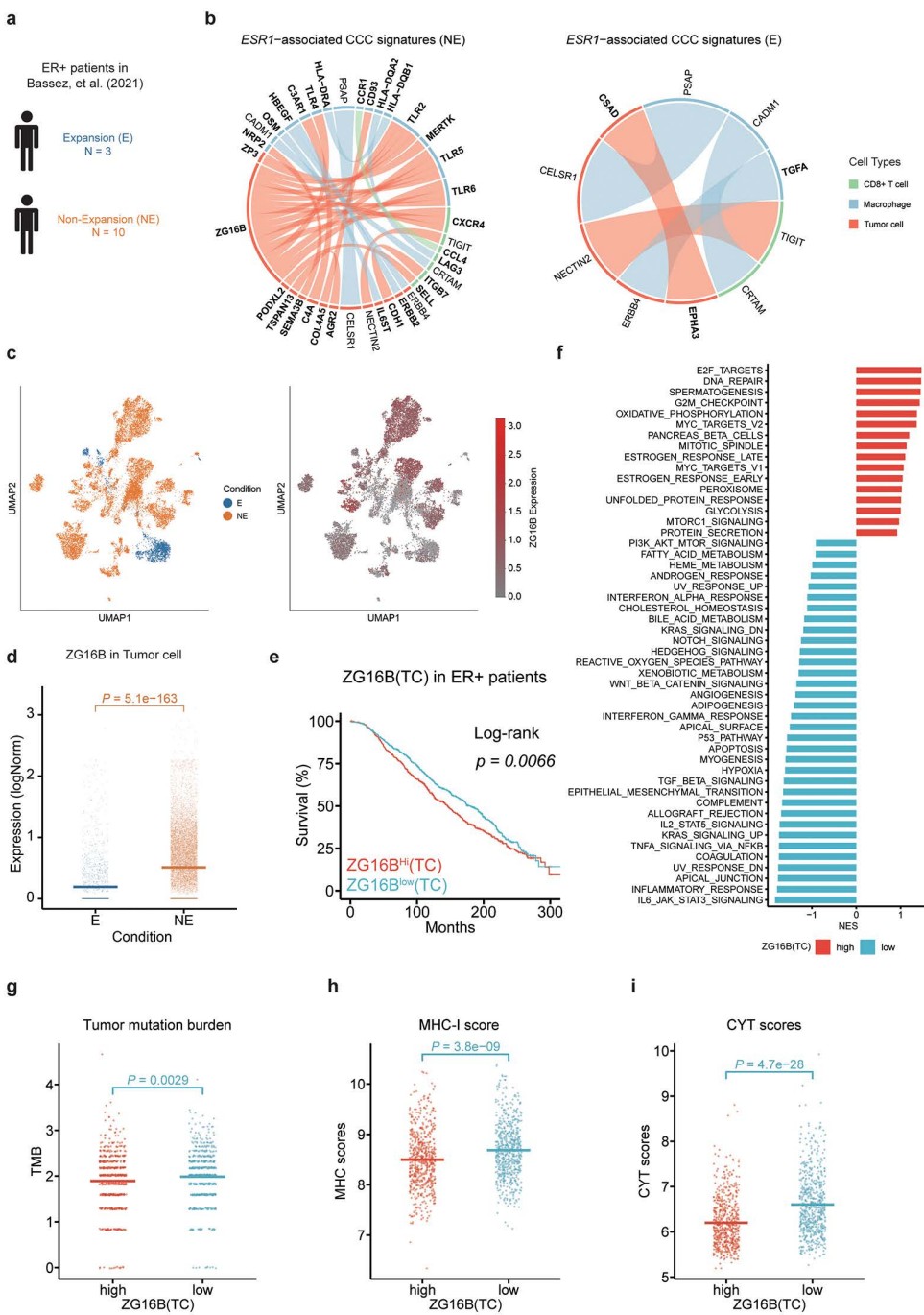

**Fig 4. *ESR1*-associated CCC signatures are linked to anti-PD1 therapy responses in breast cancer. a** Schematic representation of ER+ patients from the scRNA-seq dataset published by Bassez et al. [54].**b** Chord diagram visualization of *ESR1*-associated CCC signatures in two group of patients. Node represent genes, and colors indicate cell types: red for tumor cells, blue for macrophages, and green for CD8+ T cells. Edge colors indicates the sender cell type that expresses the gene. Edge width reflected the association strength between cancer driver and corresponding driver gene-associated CCC signatures. It is inversely proportional to the adjusted *P* value. **c** UMAP visualization of *ZG16B* expression level across the E and NE conditions. **d** Comparison of *ZG16B* expression level between the two conditions. Each dot represents the log-normalized expression of *ZG16B* in a tumor cell; the horizontal bars represent the mean expression level within each condition. *P* values were computed using a one-sided t-test. **e** Kaplan-Meier analysis of survival outcomes in ER+ cohort in METABRIC (n = 1196 patients in total) with different *ZG16B* expression level. TC, Tumor Cell. **f** GSEA analysis of differentially expressed genes between high and low *ZG16B*(TC) expression groups in the ER+ cohort, showing hallmark gene sets. **g-i** Evaluation of Tumor mutation burden **(g)**, MHC-I class molecular expression level using MHC-I scores **(h)** and cytolytic activity using CYT scores **(i)** across *ZG16B*(TC) high and low expression groups in the ER+ cohort. *P* values were computed using a one-sided t-test.

PLOS Computational Biology | https://doi.org/10.1371/journal.pcbi.1013973   February 17, 2026

was linked to unfavorable clinical outcomes (Fig 4e). We further investigated the functional states of the METABRIC cohort with different *ZG16B*(TC) expression levels using gene set enrichment analysis (GSEA) (Fig 4f) [60]. The results showed that cell proliferation-related functions such as E2F_TARGETS, G2M_CHECKPOINT, and DNA_REPAIR, were differentially enriched in patients with high *ZG16B*(TC) expression levels. Notably, ESTROGEN_RESPONSE_LATE and ESTROGEN_RESPONSE_EARLY were both enriched in this group, further demonstrating the correlation between *ZG16B* and *ESR1*. In contrast, more immune-related pathways were enriched in the low *ZG16B*(TC) expression group indicating higher immune activity in this group compared to the high-expression group. Furthermore, this phenomenon is consisted with the enriched functions of the *ESR1*-associated CCC signatures we identified. We further explored the immune status of the two *ZG16B*-stratified groups using immune-related scores (Fig 4g-4i). Tumor mutation burden (TMB), defined as the total number of somatic non-synonymous mutations present within the cancer genome, is a predictive biomarker for immune-checkpoint inhibition benefits across cancer types [61]. The MHC-I score reflects that the activation level of the MHC-I antigen presentation pathway [62] and the CYT score depicts the cytolytic activity in the TME [63]. These three metrics have been reported related to the response of immunotherapy [64]. As expected, we found that TMB, MHC-I score, and CYT score were significantly higher in the low *ZG16B*(TC) expression group compared to the other one, indicating that the low *ZG16B*(TC) is likely to have a greater response to immunotherapy. These results further supported the potential of *ZG16B* as a biomarker for predicting the immunotherapy response of ER+ breast cancer patients.

In summary, we applied Driver2Comm to identify two groups of *ESR1*-associated CCC signatures for the NE and E patients respectively, and found that the two groups of signatures reflected the distinct immune statuses of the TME. Furthermore, we identified *ZG16B*, a potential marker gene, whose high expression is associated with poor response to anti-PD1 therapy and unfavorable clinical outcome of ER+ patients. These results demonstrated that Driver2Comm can characterize distinct CCC signatures of patients with different conditions, helping users discover condition-specific CCC signatures or genes, and providing deeper biological insights.

### Triple-negative breast cancer subtype-associated CCC signatures are linked to patient prognosis

Driver2Comm can also be used to identify CCC signatures associated with a specific cancer subtype without known driver genes. Triple-negative breast cancer (TNBC) is defined by the absence of ER, PR, and HER2 expression. We extended the application of Driver2Comm to the TNBC single-cell transcriptomics data and identified 21 TNBC-associated CCC signatures. Among these signatures, we identified five immune-checkpoint targeting genes curated in CKTTD [65], including *CCR5*, *CTLA4*, *CD274*, *PDCD1*, and *VEGFA*. Then, we conducted survival analysis on TNBC-associated CCC signatures using bulk transcriptomics and clinical information data of the METABRIC TNBC patients. Among these signatures, we found that *PDCDLG2* (Mφ) − *PDCD1* (CTL) and *CD274* (Mφ) − *PDCD1* (CTL) exhibit significant prognostic effects (Fig M in S1 Text). Notably, we observed that patients bearing TNBC with *CD274* (Mφ) and *PDCD1* (CTL) activation were likely to experience significantly improved clinical outcomes (Fig 5a). Although *CD274* − *PDCD1* activation was commonly considered as an exhaustion marker of many cancer types [7], our observation was still in line with several recent studies, which reported that the expression of *PDCD1* on effector T cells was associated with enhanced cytotoxic T cell activity and anti-tumor responses in TNBC and also in other cancer types [66–68]. *CD274* and *PDCD1* overexpression at either the mRNA or the protein level was also shown to be linked with better survival of patients with TNBC [69,70].

To further investigate the potential mechanism of *CD274* (Mφ) − *PDCD1* (CTL) activation leading to favorable outcomes in the TNBC patients, we first divided the METABRIC TNBC cohort into two subgroups, including patients with both high expression levels of *CD274* and *PDCD1* (named subgroup-1) and the rest of the TNBC patients (named subgroup-2), respectively. Then, we evaluated the immunity activation status in the TME of the two TNBC subgroups by measuring the activation level of the MHC-I antigen presentation pathway using the MHC-I score [62] and the cytolytic activity using the

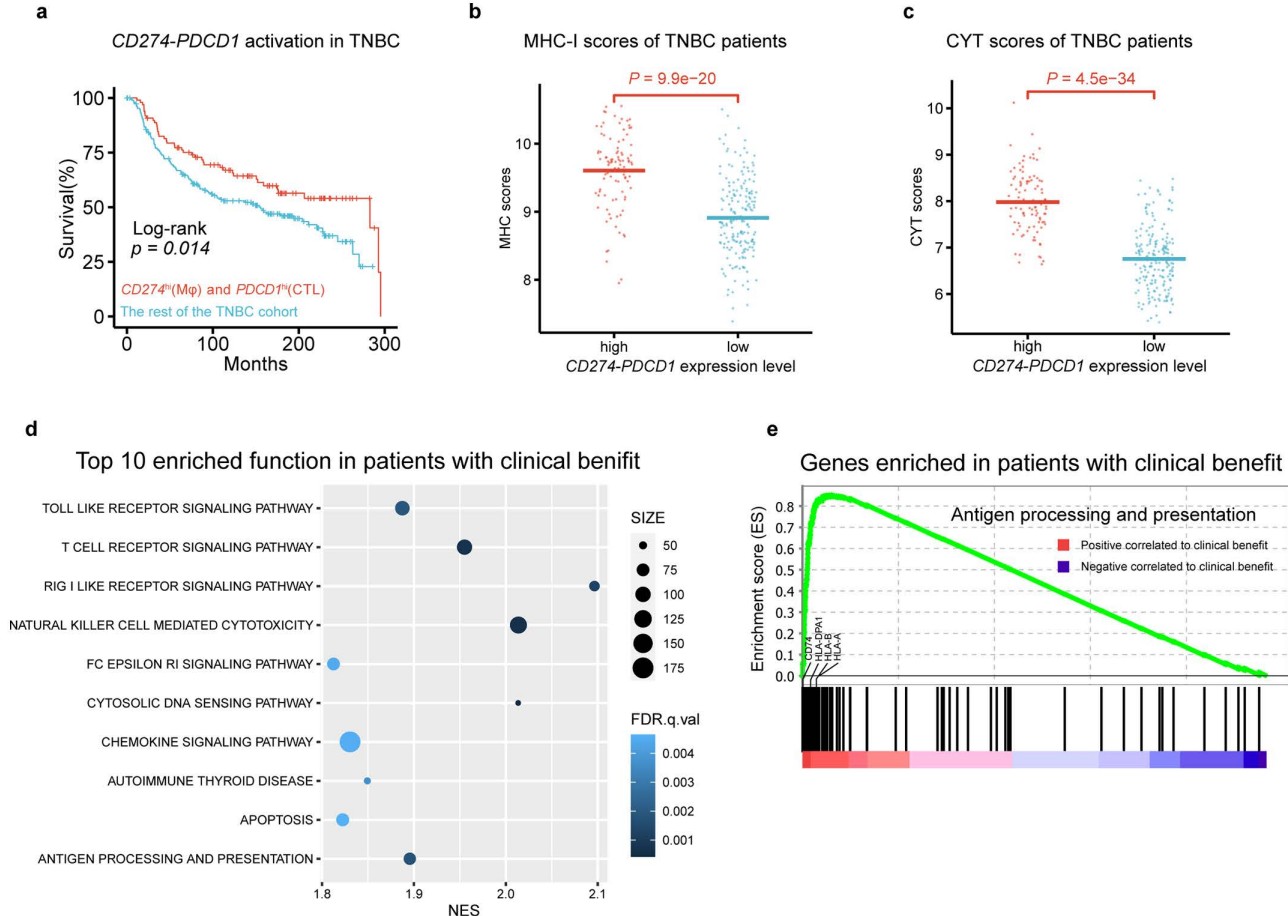

**Fig 5. Downstream analysis of TNBC-associated CCC signature 2. a** Kaplan-Meier analysis of overall survival outcomes in the TNBC patients in METABRIC (n = 290 patients in total) with different expression level of CD274 (Mφ) and *PDCD1* (CTL) (TNBC-associated CCC signature 2). P value for comparing clinical outcomes was computed using log-rank test. **b-c** MHC-I class molecular expression level evaluation using MHC-I scores (**b**) and cytolytic activity evaluation using CYT scores (**c**) of different expression level of *CD274* (Mφ) and *PDCD1* (CTL) in TNBC cohort. P values were computed using Wilcoxon rank sum test. **d** Top 10 biological function and pathways enriched in high expression level of *CD274* (Mφ) and *PDCD1* (CTL) in TNBC cohort from KEGG database identified using GSEA. Size of each dot is proportional to size of each biological functional gene set. Shades of each dot are inversely proportional to the q values which is a widely used statistical method for estimating false discovery rate (FDR). **e** Enrichment plots from gene set enrichment analysis for the antigen processing and presentation of patients in high expression level of *CD274* (Mφ) and *PDCD1* (CTL) in TNBC cohort. CTL, CD8+ T cell; Mφ, Macrophage.

CYT score [63] (Methods). We found significant increase in both MHC-I scores (one-sided *t*-test p = 9.9e-20) and CYT scores (one-sided *t*-test p = 4.5e-34) in the subgroup-1, compared to subgroup-2 (Fig 5b and 5c). Furthermore, we compared the gene expression profiles of the two TNBC subgroups using the GSEA [60] and found that genes associated with immune-related functions and pathways were up-regulated in the subgroup-1 (Fig 5d). These immune-related functions and pathways included apoptosis, antigen processing and presentation, natural killer cell mediated cytotoxicity, T cell receptor signaling pathway, RIG-I-like receptor signaling pathway, and chemokine signaling pathway. Besides *CD274* (Mφ) – *PDCD1* (CTL), we observed that genes in the remaining TNBC-associated CCC signatures, such as *CD74*, *HLA-DPA1*, *HLA-A*, and *HLA-B,* were also enriched in immune-related functions such as antigen processing and presentation (Fig 5e). In summary, these results demonstrated the capacity of TNBC-associated CCC signatures to provide insights into TNBC-specific immunity patterns.

### *MYCN*-associated CCC signatures exhibit cell-type-pair-specific functional modularity

In previous analyses, we focused on the interactions among cancer cells, macrophages, and CD8+T cells, as macrophages and CD8+T cells are known to have close interactions and play central roles in innate and adaptive immunity, respectively [71,72]. To evaluate the applicability of Driver2Comm beyond canonical immune interactions in the TME, we applied it to a neuroblastoma dataset from Yu et al. [73], comprising 22 patients, eight major cell types, and annotations for *ALK* and *MYCN* driver status. We first constructed the MCTC network among all eight major cell types and then used Driver2Comm to identify driver gene-associated CCC signatures. As a result, we identified 29 and 1 *MYCN*- and *ALK*-associated CCC signatures, respectively (Fig N in S1 Text). We focused on *MYCN*-associated CCC signatures for the subsequent analysis.

Among all cell-type pairs, fibroblast-to-endothelial communication represented the dominant *MYCN*-associated CCC signatures. As shown in Fig 6a, the key CCC signatures included *ANGPT1 − TEK*, *ANGPT1 − TIE1*, *COL18A1 − ITGA5*, and *COL18A1 − KDR*. Functional enrichment analysis revealed that these signatures were strongly enriched for angiogenesis-related processes, such as blood vessel morphogenesis, tube morphogenesis, blood vessel development, and cell-substrate adhesion (Fig 6b). This aligned with previous findings linking *MYCN* amplification to increased vascularity in neuroblastoma [74–76] and suggested that fibroblasts may serve as mediators of *MYCN*-driven vascular remodeling.

Beyond angiogenesis-related pathways, *MYCN*-associated CCC signatures revealed distinct immune regulatory modules. Specifically, endothelial-to-macrophage interactions including *APP − CD74*, *APP − LRP1*, *A2M − LRP1*, and *ADAM15 − ITGA9* were enriched for pathways involved in immune cell migration, locomotion, and immune system processes (Fig 6b), consistent with the known role of endothelial cells in myeloid cell recruitment and positioning within the TME [77,78]. By contrast, *MYCN*-associated CCC signatures involving T cells were relatively sparse across cell-type pairs (Fig 6a). Only limited fibroblast-to-T cell and macrophage-to-T cell CCC signatures were observed. These signatures were primarily linked to adhesion and vesicle-mediated processes (Fig 6b), which may reflect immunologically "cold" status in the TME in neuroblastoma with *MYCN* amplification [79–81].

In addition, *MYCN*-associated CCC signatures involving fibroblast-to-endothelial interactions such as *EFNB2 − EPHA6* were enriched for the ephrin receptor signaling pathway (Fig 6b), which is well known for its roles in neural and vascular development. Prior studies have linked ephrin signaling components with *MYCN* amplification and prognosis in neuroblastoma [82,83].

Taken together, these results demonstrate that *MYCN*-associated CCC signatures are organized into distinct, cell-type-pair-specific functional modules that collectively coordinate vascular remodeling, immune regulation, and developmental signaling within the neuroblastoma TME.

### Extension of Driver2Comm to spatial transcriptomics data

To further evaluate the applicability of Driver2Comm to spatial transcriptomics, we applied it to an HPV-negative oral squamous cell carcinoma (OSCC) dataset generated using the Visium V1 technology [84]. This dataset consists of 12 patients with detailed pathological annotations, including squamous cell carcinoma (SCC), lymphocyte negative stroma (LNS), lymphocyte positive stroma (LPS), muscle, etc. We first used SpatialinferCNV [85] to identify variated driver genes of each patient (Methods; Fig O in S1 Text). Driver2Comm was then used to systematically associated these cancer cell-intrinsic driver genes and -extrinsic CCC across all pathological structures. Driver2Comm identified CCC signatures significantly associated with *CDKN2A*, *EGFR*, *KMT2C*, *NOTCH1*, *HRAS*, and *NSD1* (Fig P in S1 Text). Similar to previous observation, the number of CCC signatures associated with different driver genes varied (Fig 7a). We focused on the two most abundant set: *CDKN2A*- and *EGFR*-associated CCC signatures for the subsequent analysis.

We first visualized the associations of *CDKN2A* and *EGFR* with their corresponding CCC signatures, as well as the top ten enriched biological functions of these signatures (Fig 7b). As shown, the key CCC signatures associated with *CDKN2A* included *APOE* (LNS) − *VLDLR* (SCC) and *ULBP2* (SCC) − *KLRK1* (LPS). These CCC signatures were

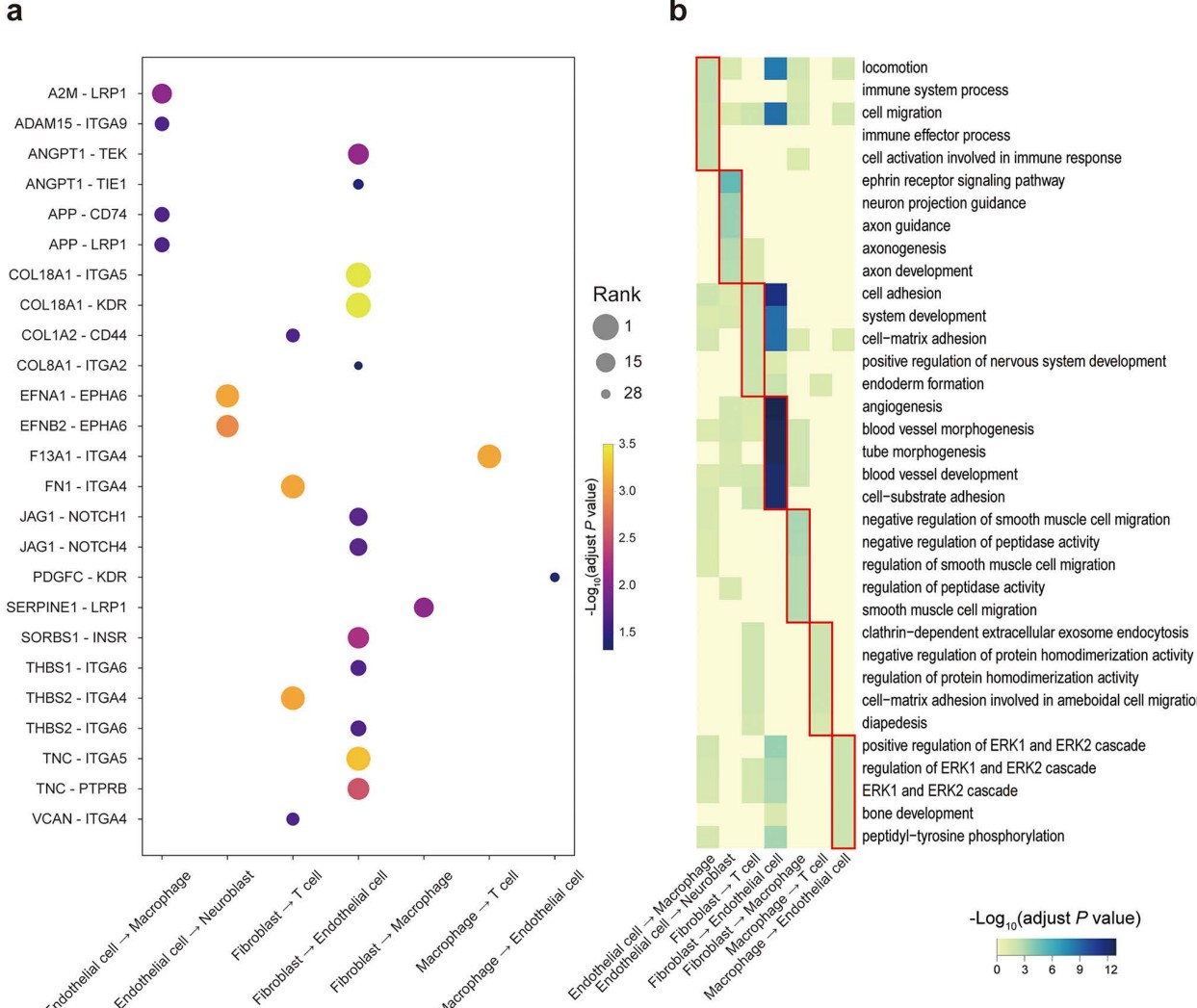

**Fig 6. Cell-type-pair-specific functional modularity of *MYCN*-associated CCC signatures in the neuroblastoma dataset. a Dot plot summarizing *MYCN*-associated CCC signatures across different cell-type pairs in the neuroblastoma dataset.** Each dot represents a ligand-receptor in CCC signatures, positioned by the corresponding sender–receiver cell-type pair. Dot size indicates the rank of the CCC signature based on association significance, while color denotes the strength of association calculated by –log₁₀(adjusted *P* value). b Heatmap showing functional enrichment of *MYCN*-associated CCC signatures stratified by cell-type pairs. Rows represent enriched Gene Ontology (GO) biological process terms, and columns represent distinct sender-receiver cell-type pairs. Shades of blue in the heatmap are inversely proportional to the enrichment *P* values that were adjusted for multiple testing using the Benjamini-Hochberg method. Nonsignificant *P* values (> 0.05) are indicated in yellow. Red outlines highlight top five enriched functional modules for each cell-type-pair CCC signatures.

enriched in angiogenesis-related processes, including blood vessel and vasculature development. These findings suggested that loss of *CDKN2A* may reshape communication between tumor and stromal compartments to promote angiogenesis, consistent with prior studies showing that *CDKN2A* and its products inhibit angiogenesis in SCC and other tumor types [86–88]. On the other hand, *EGFR* was primarily associated with *WNT5A*(LNS) – *LRP5* (SCC) and *COL1A2* (LNS) – *ITGA2* (SCC), with these signatures enriched in functions like keratinization and epithelial cell differentiation. These results implied that *EGFR* primarily influences the epithelial differentiation and keratinization programs, thereby maintaining

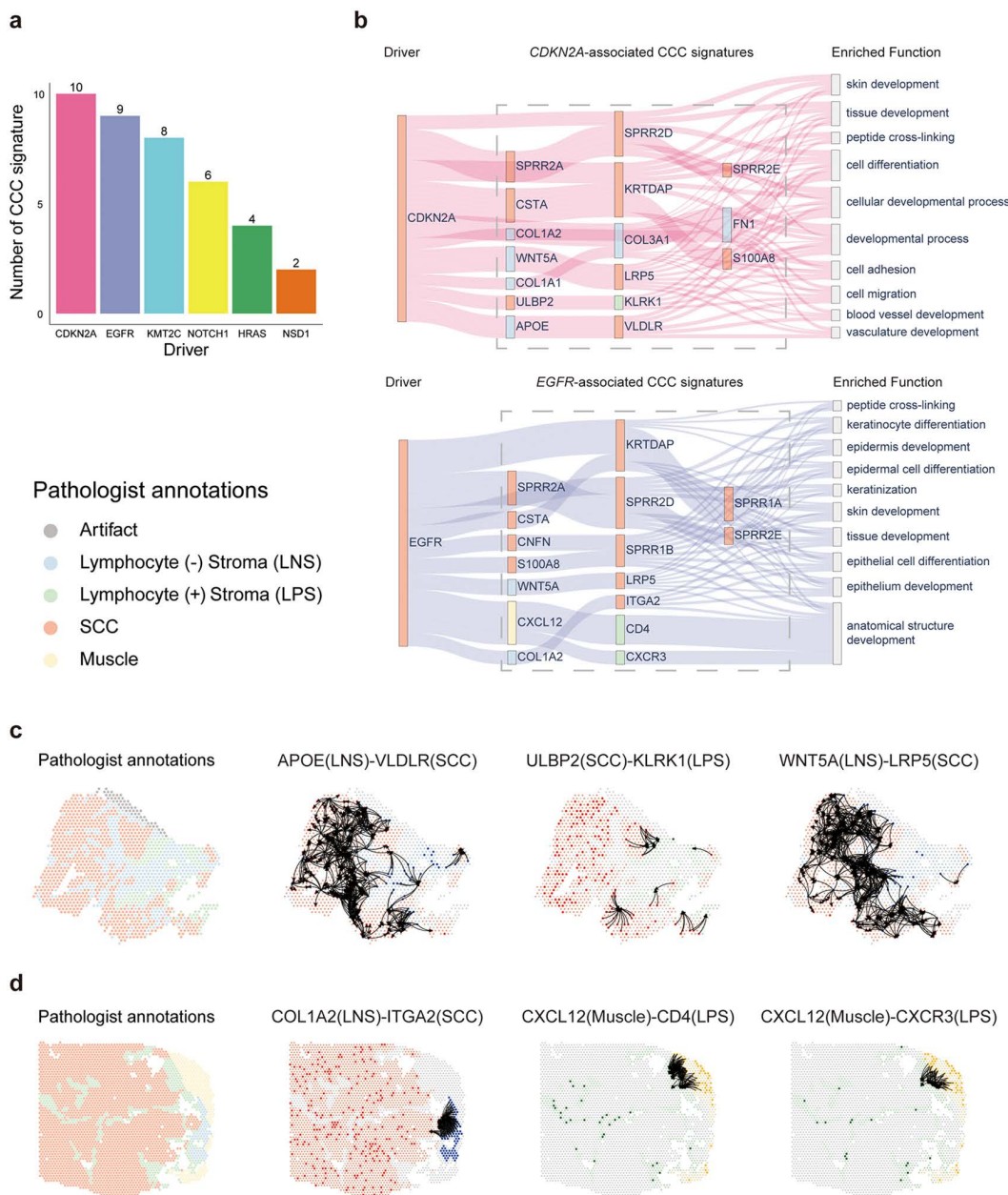

**Fig 7. Driver gene-associated CCC signatures in the OSCC dataset. a The number of CCC signatures associated with drivers in the OSCC Visium dataset. b Sankey diagrams describing the correlation among driver genes, driver gene-associated CCC signatures and the corresponding top enriched functions.** For nodes in driver and driver gene-associated CCC signatures, node labels are gene names. Node colors denote cell types: red for squamous cell carcinoma (SCC), blue for lymphocyte-negative stroma (LNS), green for lymphocyte-positive stroma (LPS), and yellow for muscle. For nodes in enriched function, node labels denote top 10 enriched function of genes in identified CCC signatures. The line width reflected the statistical significance of CCC signature enriched in the given functional category, as determined by Gene Ontology (GO) enrichment analysis (adjusted *P* value calculated using the Benjamini-Hochberg procedure). It is inversely proportional to the adjusted *P* value. c Spatial maps of a representative *CDKN2A*-mutant OSCC sample. The first panel shows pathologist annotations. The subsequent panels show the activation of representative CCC signatures identified by Driver2Comm, including *APOE*(LNS)-*VLDLR*(SCC), *ULBP2*(SCC)-*KLRK1*(LPS), and *WNT5A*(LNS)-*LRP5*(SCC). d Spatial maps of a representative *EGFR*-mutant OSCC sample. The first panel shows pathologist annotations. The subsequent panels show the activation of representative CCC signatures identified by Driver2Comm, including *COL1A2*(LNS)-*ITGA2*(SCC), *CXCL12*(Muscle)-*CD4*(LPS), and *CXCL12*(Muscle)-*CXCR3*(LPS). In all spatial maps, edges highlight potential spots engaging in the corresponding CCC interactions.

aberrant keratinization and stem-like states, and thus promoting the tumor progression [89,90]. We further visualized the representative CCC signatures. The results showed that identified CCC signatures exhibit clear co-localization regions, providing spatial support for the interactions inferred by Driver2Comm (Fig 7c and 7d).

In summary, these results demonstrate that Driver2Comm can be effectively extended to spatial transcriptomics data, enabling the identification of driver gene-associated CCC signatures that are not only functionally interpretable but also supported by spatially coherent interaction patterns within tumor tissues

## Discussion

To bridge cancer cell-intrinsic genetic variation and -extrinsic CCC in the TME, we develop Driver2Comm, a general computational framework to identify robust CCC signatures significantly associated with cancer driver genes. Unlike conventional CCC tools such as CellChat [25], which aim to infer all significant CCC patterns, or those like MultiNicheNet [28] that focus on condition-specific CCC patterns in the TME, Driver2Comm represents the first method capable of nominating complete signaling pathways linking cancer cell driver genes to their associated CCC signatures. These pathways, defined as IE pathways, provide a deeper insight into the interaction between cancer cell-intrinsic and -extrinsic factors.

In this study, we applied Driver2Comm to identify driver gene-associated CCC signatures across multiple datasets, In PDAC, we found that *BRCA1* and *AKT2*-associated CCC signatures reflected the distinct effects of the two driver genes on reshaping the TME, such as immunosuppression and cancer metastasis. In breast cancer, we identified the prognostic effects of *ESR1*-associated CCC signatures and *ZG16B* as a potential marker gene for response to anti-PD1 therapy and clinical outcomes in the ER+ subtype. We further perform a comprehensive comparison of survival relevance across driver gene-associated signatures identified by different methods, demonstrating that Driver2Comm consistently identifies CCC signatures with stronger and more consistent survival relevance (Note B in S1 Text). Extending the application of Driver2Comm to the TNBC subtype, we observed the distinct effect of *CD274 - PDCD1* interaction in the TNBC patients. In addition, we demonstrated the applicability of Driver2Comm to more complex tumor ecosystems and spatial transcriptomics data, highlighting its flexibility across the selection of cell types in the MCTC network and data modalities. Simultaneously, the robustness assessment of Drvier2Comm showed that the Driver2Comm is highly robust to moderate levels of driver gene annotation errors (Note C in S1 Text). To further validate the reproducibility of our findings, we compared the CCC signatures across independent breast cancer datasets and an integrated breast cancer dataset. The results showed a reasonable overlap in the genes and enriched functions of CCC signatures across these datasets (Fig L, Q, and R in S1 Text), confirming the reproducibility of the results. These findings not only highlighted the effectiveness and reliability of Driver2Comm in identifying driver gene-associated CCC signatures but also underscored its potential to guide personalized cancer therapy by uncovering condition-specific biomarkers and immune signatures.

Driver2Comm succeeds in associating cancer driver genes and CCC for two main reasons. First, Driver2Comm leverages the ability of CytoTalk to de novo construct CCC signaling networks that incorporate both intracellular and intercellular signaling transduction pathways, enabling the direct identification of IE pathways. Secondly, the GWAS-inspired hypothesis testing framework employed in Driver2Comm statistically prioritizes driver gene-associated CCC signatures over background noise.

From a computational perspective, the primary cost of Driver2Comm lies in the frequent subnetwork mining step, which is implemented using the efficient gSpan algorithm. Importantly, computational cost is strongly influenced by the minimum support threshold, which in practice provides a flexible handle to balance sensitivity and efficiency as cohort size increases. Another important procedure built in Driver2Comm is the association testing to link cancer cell-intrinsic and -extrinsic features. Therefore, this method may be underpowered when applied to datasets with limited samples. To tackle this problem, we used a resampling strategy to generate sufficient pseudo-samples and then employed a frequent subnetwork mining method for robust feature extraction. In the future, we aim to extend Driver2Comm to associate continuous

genetic variation such as the copy number alteration with CCC in the TME, given the fast development of computational methods for inferring copy number alteration using single-cell transcriptomics [38,91–93].

In summary, Driver2Comm enables the identification of cancer cell-extrinsic CCC signatures significantly associated with -intrinsic driver genes and explicit illustration of potential signaling pathways between them. With the rapid development of single-cell transcriptomics technologies, Driver2Comm opens up an effective way to decipher the mechanism of interactions between cancer cell-intrinsic gene variations and -extrinsic CCC events in the TME. Since targeted therapies and immunotherapies are primarily developed based on cancer cell-intrinsic genetic variations and -extrinsic CCC, respectively, Driver2Comm holds significant potential to guide the development of combination therapies of both therapeutic strategies, which could improve patient outcomes.

## Methods

### Preprocessing of single-cell transcriptomics data

For all single-cell transcriptomics data used in this study, we used Seurat [94] to normalize our data and only retained protein-coding genes based on annotations from CytoTalk [36]. Besides, we removed genes expressed in less than 5% of all cells of a given type.

### Simulation of single-cell transcriptomics data for pseudo-samples

Up-sampling is a commonly used technique to deal with imbalanced sample distribution problems or small sample issues [95]. Due to the high cost of single-cell experiments, the patient cohort in a single-cell transcriptomics experiment is relatively small. Detail information of the rules of patient selection have been shown in Note D in S1 Text. To address this issue, we performed an up-sampling procedure as below to generate sufficient pseudo-samples.

In this study, we focused on the CCC among macrophages, CD8+T cells, and cancer cells in the TME. Single-cell transcriptomics data of other cell types were used as background expression distributions to compute cell-type-specific gene expression levels used in CytoTalk [36]. Therefore, we only sampled the single-cell transcriptomics data of macrophages, CD8+T cells, and cancer cells for each patient. To ensure the effectiveness of the results in this study, we set a minimum threshold of 20 cells for each of these three cell types. For our sampling process, we randomly sampled 50% of the cells for each cell type with replacement to generate a new set of gene expression data for each pseudo-sample. This procedure was repeated five times per patient.

For data from Hwang et al., 10 patients satisfied the sampling criteria, resulting in 50 pseudo-samples. For data from Wu et al., 12 patients satisfied the sampling criteria, while patient CID45171 did not meet the criteria. However, we chose to include CID45171 as a pseudo-sample, which brought the total number of pseudo-samples in this dataset to 61. For data from Bassez et al., 25 patients satisfied the sampling criteria. BIOKEY_5, BIOKEY_7, and BIOKEY_25 did not comply with the sampling criteria and were, therefore, treated as pseudo-samples. This approach allowed us to construct a total of 128 pseudo-samples, which were used for all subsequent analyses in this dataset.

### Construction of the MCTC network

CCC in the tissue microenvironment occurs among multiple cell types simultaneously. To model the complicated interaction in the TME, we constructed an MCTC network for each patient. The MCTC network in this study includes macrophages, CD8+ T cells, and cancer cells given the critical role of macrophages and CD8+ T cells in innate and adaptive immunity against cancer.

CytoTalk is a computation tool designed for de novo construction of a signal transduction network between two different cell types [36]. Unlike the computational methods inferring CCC at the ligand-receptor pairs level, CytoTalk constructs a cell-type communication network composed of both intracellular and intercellular interactions, which enable the visualization of how the cancer cell-intrinsic factors and -extrinsic factors interact in an intact pathway.

We constructed the MCTC network in the following steps. We first constructed a signal transduction network for each two cell type pair using CytoTalk. Then we removed the node prizes and edge weights in these networks. Finally, we took the union of these signal transduction networks between two cell types to construct the MCTC network for each patient.

**Frequent subnetwork mining**

Frequent subnetwork mining identifies frequently occurring subnetworks in a given graph dataset. This method has a wide range of applications such as the detection of common substructures in sets of biomolecules and the discovery of network motifs in large-scale molecular interaction networks [96]. The fundamental hypothesis of these applications is that the conserved subnetworks in the graph collection may indicate some underlying biological relevance. In this study, we focused on the frequent subnetworks associated with cancer driver genes in the MCTC networks. Considering frequent subnetworks as features of the MCTC networks exhibits two key benefits. First, the features are less influenced by a particular sample, enhancing the robustness of the method. Second, the frequent subnetworks are more likely to represent the universal communication patterns as the number of graphs increases. We performed frequent subnetwork mining on the MCTC networks using gSpan [97], which is an efficient algorithm for frequent subnetwork mining based on depth-first search. The frequent subnetwork mining results were formulated into a 0–1 matrix using one-hot encoding. The element $f_{ij} = 1$ in matrix $F$ represented *frequent subnetwork$_j$* appearring in *patient$_i$*'s MCTC network.

The support threshold of the frequent subnetwork mining algorithm affects generated candidate CCC signatures. Choosing the threshold is a fundamental trade-off between robustness and heterogeneity. Lowering the threshold simultaneously increases the quantity and heterogeneity of candidate CCC signatures and enhances the likelihood of discovering driver gene-associated CCC signatures. However, this expansion comes at the expense of analysis efficiency. Conversely, raising the threshold enhances the robustness of CCC signatures but increases the risk of overlooking driver gene-associated CCC signatures. We recommend that users can choose support thresholds based on the proportion of patients with interested driver genes and the desired number of retained frequent subnetworks. The detail setting of parameters in this study was list in Note E in S1 Text.

**Association testing**

After conducting frequent subnetworks mining, we obtained numerous frequent subnetworks of MCTC networks as the candidate CCC signatures. To identify cancer driver gene-associated CCC signatures, we performed an association testing procedure between cancer driver genes and frequent subnetworks.

The inspiration for association testing is derived from GWAS [35]. Typically in GWAS, linear or logistic regression models are used to test for associations, depending on whether the phenotype is continuous or binary, respectively. In this study, both cancer driver genes and frequent subnetworks were encoded as 0–1 matrix using one-hot encoding. Therefore, we performed a one-sided Fisher's exact test using *scipy* [98] given that Fisher's exact test is a statistical test used to determine if there are nonrandom associations between two binary variables [99]. Besides, the reason why we used the one-sided Fisher's exact test is that we were interested in the frequent subnetworks that emerged together with a specific cancer driver gene instead of being mutually exclusive. All P values corresponding to frequent subnetworks were adjusted using the method of Benjamini-Hochberg procedure [100]. All frequent subnetworks with adjusted P value smaller than the predefined threshold were considered significant and defined as cancer driver gene-associated CCC signatures. Finally, we ranked all these significant signatures in descending order of significance of association test with the corresponding cancer driver.

**Visualization and functional enrichment analysis of IE pathways**

To further investigate the mechanism of interactions between cancer cell-intrinsic and -extrinsic factors, we identified IE pathways connecting cancer driver genes with their associated CCC signatures in the MCTC network. IE pathways were defined as the shortest signaling paths linking a driver gene to its associated CCC signatures.

We identified IE pathways using the following procedure. In the previous step, driver-associated CCC signatures were extracted from the MCTC network. Because the MCTC network was centered around ligand–receptor pairs [34], cancer driver genes were not guaranteed to be included in this network. To explicitly connect cancer cell-intrinsic driver genes with -extrinsic CCC signatures, we expanded the MCTC network to incorporate the driver gene using the CytoTalk-built intracellular network. Specifically, for each node in the MCTC network, we iteratively incorporated its first-order neighbors from the intracellular network until the driver gene was reached. Finally, the shortest paths between the driver gene and its associated CCC signatures in the expanded MCTC networks were identified as the IE pathways.

IE pathways were further visualized using Cytoscape [101] and evaluated by functional enrichment analysis using g:Profiler [102]. Redundant Gene Ontology terms were removed using REVIGO [103]. Enrichment p values were adjusted for multiple testing using the method of Benjamini-Hochberg procedure.

## Functional enrichment analysis of TNBC patients

To investigate functions of up-regulated genes in the TNBC patients with both high expression levels of *CD274* (Mφ) and *PDCD1* (CTL), compared to the rest of the TNBC METABRIC patients, we conduct GSEA [60]. Signal2Noise metric was used to rank the gene in different phenotypes. C2 and C7 gene lists from MSigDB (http://software.broadinstitute.org/gsea/msigdb/) were used for GSEA.

## Survival analysis using the TCGA and METABRIC cohort

To evaluate the prognostic effect of cancer driver gene-associated CCC signatures, we investigated the association between patient overall survival in the TCGA and METABRIC cohort and the expression level of genes in these CCC signatures. However, a major challenge of this analysis is how we can stratify the patient cohort using those CCC signatures when single-cell transcriptomics data is not available for the bulk cohorts.

To address this challenge, we converted the different expression levels of a CCC signature to the abundance of cell subtypes with different expression levels corresponding to that CCC signature. We then stratified patients by the abundance of these subtypes (Fig E in S1 Text). The conversion was based on the hypothesis that a cell type with a high expression level of a specific gene could be considered as a distinct cell subtype. Moreover, if the abundance of this subtype was relatively high, the gene was likely to be highly expressed. We used the following procedure to accomplish the conversion. For each gene in a cancer driver gene-associated CCC signature, we first used the median expression level to categorize the cell type into two subtypes using single-cell transcriptomics data. Subsequently, we estimated the abundance of the subtype with a high expression level of the gene among patients. Finally, we stratified patients based on the median split of abundance of the subtype mention above. Patients with higher abundance of this subtype would be consider as patients exhibiting higher expression level of that gene in the CCC signature.

For instance, for *PDCD1* (CTL) in TNBC-associated CCC signature 2, we stratified CD8+T cells into $PDCD1^{Hi}$-CD8+ T cells and $PDCD1^{Low}$-CD8+ T cells by the median expression level of *PDCD1* in CD8+ T cells. Secondly, we estimate the abundance of $PDCD1^{Hi}$-CD8+T cell subtype by CIBERSORTx. We then stratified patient cohorts into two groups based on the median split of the estimated abundance of $PDCD1^{Hi}$-CD8+ T cell subtype. Patients with a higher abundance of $PDCD1^{Hi}$-CD8+ T cell subtype would be considered as patients exhibiting a higher expression level of *PDCD1* (CTL) in TNBC-associated CCC signature 2. Finally, we conducted survival analysis of two groups of patients with different $PDCD1^{Hi}$-CD8+ T cell subtype abundance.

Survival curves were generated using the Kaplan–Meier method with the 'survival' package v.3.2-13. We assessed the significance between two groups using the log-rank test statistics.

## Deconvolution of bulk RNA profiles

CIBERSORTx v1.0 (online) [104] was used to generate the signature matrices of cell types in the TME and estimate cell type abundance from bulk RNA profiles. To generate the background cell type signature matrix for cell type annotation, we

 

randomly subsampled 15% of cells for each cell type. All the parameters of CIBERSORTx in this study obeyed the suggestion in Steen C B et al. [105]. (Note F in S1 Text).

## Immune-related benchmarking

To investigate the mechanism of the different clinical outcomes between the two patient groups, we evaluated the immunity activation status in the TME using two immune-related scores: the MHC-I score and the CYT score. MHC-I score measures the activity of the antigen processing and presentation by the MHC-I complex and is calculated as the mean of the "core" MHC-I set including *HLA-A, HLA-B, HLA-C, TAP1, TAP2, NLRC5, PSMB9, PSMB8,* and *B2M*. [62] CYT score measures the cytolytic activity and is calculated as the geometric mean of two key cytolytic effectors including *GZMA* and *PRF1*. These two effectors will dramatically be upregulated upon CD8$^+$ T cell activation [63].

## Cancer driver gene identification

In this study, we identified driver genes affected by copy number variation (CNV) in the PDAC and OSCC dataset using the following procedure. First, for each patient, we inferred the CNV profile of each gene in tumor cells and reference diploid cells using inferCNV [38] (Fig A in S1 Text), which is designed to infer CNV from single-cell transcriptomic datasets. For the spatial transcriptomics dataset, we applied SpatialInferCNV [85], an extension of inferCNV for spatial transcriptomics data. Next, we curated a list of known oncogenes and tumor suppressor genes from previous studies as the candidate driver genes. (Note G in S1 Text). For each candidate driver gene in each patient, we extracted its inferred CNV profile in the tumor cells as the query CNV distribution. Simultaneously, we generated the reference CNV distribution for this candidate driver gene by extracting its CNV profile from reference diploid cells. Finally, we used the Kolmogorov–Smirnov (KS) test to assess if there were significant differences between the query CNV distribution and the corresponding reference CNV distribution of each candidate driver gene in each patient. P values were adjusted for multiple testing using the Benjamini-Hochberg method. For each patient, candidate genes with adjusted p values of less than 0.01 were identified as driver genes. All inferCNV parameters were set to default values except for *window_length*, which was set to 25 to obtain a finer resolution CNV profile. For the PDAC dataset, immune cell types including T cells, B cells, and natural killer cells were used as reference diploid cells. For the OSCC dataset, pathological annotations including non-cancerous mucosa, lymphocyte positive stroma, and keratin were used as reference diploid compartments.

In the PDAC dataset, we identified that *KRAS*, *BRCA1*, *AKT2*, *CDKN2A*, *GATA6*, *PTEN*, and *MET* were variated. Since *KRAS* were variated in all samples in this study, Driver2Comm could not identify specific information, so *KRAS* was not considered in subsequent analysis.

In the OSCC dataset, we identified that *CDKN2A, FAT1, NOTCH1, HRAS, CCND1, EGFR, KMT2C,* and *NSD1* were variated.

## Running of published CCC inference method

To systematically evaluate how different CCC inference methods influence Driver2Comm, we applied an established CCC inference method, scMLnet [34], which models both intercellular communication and intracellular downstream regulatory responses.

The R package scMLnet (v0.1.0) was applied to the PDAC dataset and the breast cancer dataset from Wu et al. (Note B in S1 Text), following the workflow recommended in the official tutorial (https://github.com/SunXQlab/scMLnet/blob/master/vignettes/Tutorial_of_scMLnet.md). All analyses were performed using the default parameters.

## Dataset integration

To further demonstrate the applicability of Driver2Comm to datasets integrated from different sources, we combined two independent breast cancer datasets into an integrated dataset for analysis. Single-cell transcriptomic data from different sources often contain differences in capturing times, handling personnel, reagent lots, equipment, and even technology

platforms [106]. These differences, known as batch effects, may confound our analysis. Thus, we corrected the batch effect of these two datasets using our previously developed method Beaconet [107] (Fig R in S1 Text), which serves as an effective and efficient batch effect removal tool that can facilitate the integration of single-cell datasets in a reference-free and molecular feature-preserved mode.

Specifically, we took all the cells belonging to the primary untreated samples in both datasets and corrected their batch effects with all the overlap genes. As a result, we obtained an integrated dataset consisting of 41 patients, a total of 146,031 cells, and 19,093 genes.

## Code availability

A Python implementation for Driver2Comm and example scripts for associating driver genes of cancer cells with CCC in the TME using single-cell transcriptomics data are available at https://github.com/huBioinfo/Driver2Comm.

## Supporting information

**S1 Text. Supplementary material. Text document with results, figures, tables, and descriptions of additional experiments.**
(DOCX)

## Acknowledgments

We thank all members of Prof. Gao's lab at Xidian University for fruitful discussion.

## Author contributions

**Conceptualization:** Runzhi Xie, Yuxuan Hu, Lin Gao.

**Data curation:** Runzhi Xie, Yuxuan Hu.

**Formal analysis:** Runzhi Xie, Yuxuan Hu.

**Funding acquisition:** Yuxuan Hu, Lin Gao.

**Investigation:** Runzhi Xie, Yuxuan Hu.

**Methodology:** Runzhi Xie, Junping Li, Yuxuan Hu.

**Project administration:** Yuxuan Hu, Lin Gao.

**Resources:** Yuxuan Hu, Lin Gao.

**Software:** Runzhi Xie.

**Supervision:** Yuxuan Hu, Lin Gao.

**Validation:** Runzhi Xie, Junping Li, Yuxuan Hu.

**Visualization:** Runzhi Xie, Junping Li, Yuxuan Hu.

**Writing – original draft:** Runzhi Xie.

**Writing – review & editing:** Runzhi Xie, Junping Li, Yuxuan Hu, Lin Gao.

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
