## [Decision Letter · Decision Letter 0]

23 Jul 2025

Bridging cancer cell-intrinsic driver genes and -extrinsic cell-cell communication with Driver2Comm

PLOS Computational Biology

Dear Dr. Gao,

Thank you for submitting your manuscript to PLOS Computational Biology. After careful consideration, we feel that it has merit but does not fully meet PLOS Computational Biology's publication criteria as it currently stands. Therefore, we invite you to submit a revised version of the manuscript that addresses the points raised during the review process.

Please submit your revised manuscript within 60 days Sep 22 2025 11:59PM. If you will need more time than this to complete your revisions, please reply to this message or contact the journal office at ploscompbiol@plos.org. Please include the following items when submitting your revised manuscript:

We look forward to receiving your revised manuscript.

Kind regards,

Rodrigo Mora

Academic Editor

PLOS Computational Biology

Pedro Mendes

Section Editor

PLOS Computational Biology

**Additional Editor Comments:**

Please refer to the comments of the reviewers for the submission of a new version of the manuscript.

**Journal Requirements:**

1) Please provide an Author Summary. This should appear in your manuscript between the Abstract (if applicable) and the Introduction, and should be 150-200 words long. The aim should be to make your findings accessible to a wide audience that includes both scientists and non-scientists. Sample summaries can be found on our website under Submission Guidelines:

2) Please amend your detailed Financial Disclosure statement. This is published with the article. It must therefore be completed in full sentences and contain the exact wording you wish to be published. Please ensure that the funders and grant numbers match between the Financial Disclosure field and the Funding Information tab in your submission form. Note that the funders must be provided in the same order in both places as well.

**Reviewers' comments:**

Reviewer's Responses to Questions

**Comments to the Authors:**

Reviewer #1: This manuscript presents Driver2Comm, a novel computational framework that systematically bridges cancer cell-intrinsic driver genes and extrinsic cell-cell communication (CCC) events in the tumor microenvironment (TME). The study addresses a critical gap in cancer biology by integrating single-cell transcriptomics, functional pathway analysis, and clinical validation to uncover intrinsic-extrinsic (IE) pathways with remarkable biological and translational relevance.

Driver2Comm is the first tool to explicitly link driver genes to CCC signatures via de novo reconstructed multi-cell-type communication (MCTC) networks, leveraging CytoTalk for intracellular/intercellular signaling and GWAS-inspired statistical association testing. This approach elegantly overcomes limitations of existing tools. The application to three distinct cancer types (PDAC, ER+/HER2+ breast cancer, TNBC) demonstrates broad utility. Key findings—such as BRCA1-mediated immunosuppression in PDAC, ER1-associated CCC signatures predicting anti-PD1 response, and the paradoxical prognostic role of CD274-PDCD1 in TNBC—are robustly supported by survival analysis, immune scoring, and independent cohort validation.

Minor suggestions:

1. Please clarify the rationale for selecting macrophages and CD8+ T cells as focal immune populations, given the potential contribution of other stromal cells (e.g., fibroblasts, dendritic cells).

2. Need to briefly discuss computational scalability for larger datasets or additional cancer types.

Overall, Driver2Comm represents an important methodological advance with biological and clinical implications. The work provides a versatile resource for decoding cancer-immune crosstalk. I strongly recommend publication pending minor revisions.

Reviewer #2: In this manuscript, the authors propose a software to analyze the associations between cancer driver genes and cell-cell communications. The idea is interesting and the examples are impressing. However, three major points should be addressed before acceptance for publication.

First, it is an importnat but hard work to identify cancer driver mutations. The authors should explicitly assess how errors in identifying cancer driver genes affect the analytical results.

Second, as spatial transcriptomics data are accomulating for tumor samples, the authors should discuss and show how the software is applied to cancer spatial transcriptomics data. It is better to use cancer spatial transcriptomics data to validate the predictions.

Third, it is better to discuss whether cell-cell communication is helpful for identifying cancer driver genes.

Reviewer #3: Understanding how the cancer cell intrinsic driver genes interact with -extrinsic cell-cell communication (CCC) events in the TME is important for studying the cancer progression and identifying microenvironment-related targets. The authors present a computational framework Driver2Comm to identify intrinsic-extrinsic (IE) pathways that functionally connect cancer cell driver genes with their associated CCC signatures. This framework integrates the authors’ previously developed CytoTalk, frequent subnetwork mining and association testing to identify cancer driver-associated CCC signatures. The authors demonstrated the performance of Driver2Comm using three distinct cancer datasets. Below are specific comments.

1. There are several CCC inference methods that consider both intercellular CCC and intracellular downstream response, such as scMLnet and CellCall. Are these methods applicable to identify cancer driver-associated CCC signatures with the help of association testing, which can be also used to perform the survival analysis. It will be great to clarify the differences in biological discovery between Driver2Comm and other CCC methods.

2. How did the frequent subnetwork mining method ensure that the identified subnetwork consists of both intrinsic driver genes and extrinsic CCC?

3. It is important to compare the performance with the baseline methods such as simply using the cancer drivers or DEGs. In addition, the difference among the different lines in Figure 3 does not seem significant. The authors should add more descriptions of the figure legends. For example, what is the meaning of different colors in Figure 3f? How did the low-rank test perform? Please clarify the comparison groups for the indicated p-value?

4. It is not very common to study CCC among only two cell types? It is better if the authors can show one example with common number of cell types in the dataset.

5. Please clarify the meaning of the line width between CCC signatures and the enriched functions in the Sankey diagram.

**Have the authors made all data and (if applicable) computational code underlying the findings in their manuscript fully available?**

Reviewer #1: Yes

Reviewer #2: Yes

Reviewer #3: Yes

PLOS authors have the option to publish the peer review history of their article (what does this mean? ). If published, this will include your full peer review and any attached files.

**Do you want your identity to be public for this peer review?** For information about this choice, including consent withdrawal, please see our Privacy Policy .

Reviewer #1: No

Reviewer #2: No

Reviewer #3: No

**Figure resubmission:**

**Reproducibility:**



---

## [Decision Letter · Decision Letter 1]

3 Feb 2026

Dear Dr. Gao,

We are pleased to inform you that your manuscript 'Bridging cancer cell-intrinsic driver genes and -extrinsic cell-cell communication with Driver2Comm' has been provisionally accepted for publication in PLOS Computational Biology.

Best regards,

Rodrigo Mora

Academic Editor

PLOS Computational Biology

Pedro Mendes

Section Editor

PLOS Computational Biology

Reviewer's Responses to Questions

**Comments to the Authors:**

Reviewer #1: No further question.

Reviewer #3: The authors have well addressed my comments. No further comments.

**Have the authors made all data and (if applicable) computational code underlying the findings in their manuscript fully available?**

Reviewer #1: Yes

Reviewer #3: Yes

PLOS authors have the option to publish the peer review history of their article (what does this mean? ). If published, this will include your full peer review and any attached files.

**Do you want your identity to be public for this peer review?** For information about this choice, including consent withdrawal, please see our Privacy Policy .

Reviewer #1: No

Reviewer #3: No

---

## [Editor Report · Acceptance letter]

PCOMPBIOL-D-25-01112R1

Bridging cancer cell-intrinsic driver genes and -extrinsic cell-cell communication with Driver2Comm

Dear Dr Gao,

I am pleased to inform you that your manuscript has been formally accepted for publication in PLOS Computational Biology. Your manuscript is now with our production department and you will be notified of the publication date in due course.

With kind regards,

Judit Kozma
